# Evolution and origin of sliding clamp in bacteria, archaea and eukarya

**Sandesh Acharya, Amol Dahal, Hitesh Kumar Bhattarai** (ID) *

Department of Biotechnology, Kathmandu University, Dhulikhel, Nepal

* hitesh321@gmail.com

## Abstract

The replication of DNA is an essential process in all domains of life. A protein often involved in replication is the sliding clamp. The sliding clamp encircles the DNA and helps replicative polymerase stay attached to the replication machinery increasing the processivity of the polymerase. In eukaryotes and archaea, the sliding clamp is called the Proliferating Cell Nuclear Antigen (PCNA) and consists of two domains. This PCNA forms a trimer encircling the DNA as a hexamer. In bacteria, the structure of the sliding clamp is highly conserved, but the protein itself, called beta clamp, contains three domains, which dimerize to form a hexamer. The bulk of literature touts a conservation of the structure of the sliding clamp, but fails to recognize the conservation of protein sequence among sliding clamps. In this paper, we have used PSI blast to the second iteration in NCBI to show a statistically significant sequence homology between *Pyrococcus furiosus* PCNA and *Kallipyga gabonensis* beta clamp. The last two domains of beta clamp align with the two domains of PCNA. This homology data demonstrates that PCNA and beta clamp arose from a common ancestor. In this paper, we have further used beta clamp and PCNA sequences from diverse bacteria, archaea and eukarya to build maximum likelihood phylogenetic tree. Most, but not all, species in different domains of life harbor one sliding clamp from vertical inheritance. Some of these species that have two or more sliding clamps have acquired them from gene duplication or horizontal gene transfer events.

## Introduction

Replication, transcription and translation are fundamental events occurring in all living organisms. Homologous machineries exist for the transcription and translation pathways of all cells however replication machineries vary considerably among the different domains, mainly between bacteria and archaea/eukarya [1]. DNA polymerases, primases and helicases involved in replication process are quite distinct and non-homologous between bacteria and archaea/eukarya [1]. Interestingly two replication proteins, DNA sliding clamps and clamp loader share structural homology in the different domains of life.

DNA sliding clamp proteins, found in all organisms, are essential components of DNA polymerase enzyme as they bind the enzyme and prevent it from dissociating from the template DNA strand. The sliding clamp proteins are called beta clamp in bacteria and

**Data Availability Statement:** All relevant data within manuscript and its supporting information files.

**Funding:** There is no funding source for this work.

**Competing interests:** The authors declare no competing interests.

proliferating cell nuclear antigen (PCNA) in eukaryotes [2]. Since the replication machinery of archaea demonstrates similarity to eukaryotic replication machinery than to bacterial replication machinery, archaea contain PCNA and not beta clamp. While beta clamp is a homo-dimer consisting two subunits of three domains [3], PCNA is a homo-trimer made up of three subunits of two domains [4]. In some archaea of Crenarchaeal sub-domain, three PCNA homologs are present, which exist as monomers in solution and self-assemble in to a functional hetero-trimer [5, 6]. Despite the difference in the subunits and domains, all sliding clamps assemble to form a ring-shaped protein, such that the ring is wide enough to accommodate double-stranded DNA [7, 8]. The architecture consists of an outer negatively charged surface of continuous beta-sheets, with positively charged inner surface composed of alpha helices [7, 9]. In spite of sub-unit compositions, the 3-D structure of the sliding clamp has been highly conserved throughout evolution.

Sliding clamps encircle DNA and thereby, physically tether the DNA polymerase to the DNA template, hence increasing the processivity of the enzyme [10]. Previously, the function of sliding clamps was thought to be limited only to maintain the processivity of DNA polymerase. However, with recent studies, various interactions of sliding clamps with other proteins have been recognized that shed light on different roles of sliding clamps in the cell. PCNA has been known to interact with several proteins like XPG, MSH3, MSH6, MCH1, PMS2, hMYH, Fen1 endonuclease, and DNA ligase I suggesting roles in nucleotide excision repair, mismatch repair, base excision repair, and maturation of Okazaki fragments [11]. Similarly, the prokaryotic homolog of PCNA, viz. beta-clamp, has been observed to interact with MutS, PolI, DNA ligase, PolII and PolV suggesting roles in mismatch repair, processing of Okazaki fragments, and DNA repair [12, 13]. Thus, not only the 3-D structure, but also interactions and functions are conserved between sliding clamps of prokaryotes and eukaryotes.

The existing body of information reveals that bacterial beta clamps and eukaryotic PCNA share structural homology. It has been reported that beta clamps and PCNA share no sequence homology at all although beta clamp proteins and PCNA are highly conserved in bacteria and in archaea/eukaryotes, respectively [2, 3, 9]. In this study, we analyzed sequence homology between bacterial beta clamp and archaeal/eukaryotic PCNA. Additionally, to understand their evolution, sliding clamp sequences from representative bacteria, archaea and eukarya were retrieved and a phylogenetic tree was drawn for each domain of life. As expected, the phylogenetic trees largely follow the vertical evolution pattern of organisms. In some instances, they reveal gene duplication and horizontal gene transfer events.

## Material and methods

### Determination of sequence homology between bacterial beta clamp and archaeal PCNA

Two hundred forty-nine amino acid long *Pyrococcus furiosus* PCNA sequence (Uniprot id: O73947) was obtained from the Uniprot website. To discover homologues of the sequence in bacteria, the sequence was protein blasted against bacteria in the NCBI database (taxID: 2) using the NCBI web server. The non-redundant protein sequence database was used, and position specific iterated blast (PSI-Blast) was carried out using the default BLOSUM 62 Matrix, gap cost of Existence:11 Extension:1, with conditional compositional score matrix adjustment at the threshold of 0.005. To discover more distant homologues, a second iteration of PSI-blast was carried out.

The first iteration of PSI blast is identical as the normal BLASTp. Then, a multiple alignment of the highest scoring pairs of the PSI-Blast first iteration (or BLASTp) with e-value greater than 0.005 is generated. This multiple alignment is used to calculate a Position-specific

score matrix (PSSM). The newly generated PSSM is then used in place of BLOSUM62 as a substitution matrix for the second iteration of PSI-BLAST [14]. These steps are carried out automatically by the NCBI web server.

From the second iteration of the PSI blast, a number of bacterial beta clamps were found as homologues of *P. furiosus* PCNA. Specifically, domain 2 of bacterial beta clamp was found homologous to *P. furiosus* PCNA domain 1 and domain 3 of bacterial beta clamp was found homologous to *P. furiosus* PCNA domain 2. *Kallipyga gabonensis* beta clamp was found to be the most homologous bacterial beta clamp. To demonstrate the homology between *K. gabonensis* beta clamp and *P. furiosus* PCNA, domain 2 and 3 of *K gabonensis* beta clamp (starting amino acid 128) as discovered by pfam website was aligned with *P. furiosus* PCNA using clustal omega software. Similarly, human PCNA and *P. furiosus* PCNA, and domain 2 and 3 of *K. gabonensis* beta clamp and *E. coli* beta clamp (starting amino acid 129) were aligned using clustal omega. Earlier papers have reported that there is no homology between PCNA and beta clamp. To evaluate this point, human PCNA was aligned with domain 2 and 3 of *E. coli* beta clamp.

The above search was conducted on an earlier date. The same search today (December 24, 2020) reveals different homologues of PFU PCNA in bacterial domain, possibly because the database size of NCBI has increased. *Oscillatoria nigro-viridis*, *Oscillatoriales cyanobacterium USR001 and Blastocatellia bacterium* are the most significant hits with Query Cover above 95%, e-value less than 0.004, and percentage identity above 18% (18.22%- 20.08%). The full blast search results conducted on December 24, 2020 are available here. A random search of homologues of *Homo sapiens* PCNA in bacteria also gives some homologues in the second iteration of PSI-Blast, results made available here. The point is that a random blast of PCNA in the bacterial domain returns beta-clamps as significant hits in the second iteration of PSI-Blast. This suggests some sort of sequence homology between beta clamps and PCNA proteins.

## Sequence retrieval and multiple sequence alignment for tree construction

In order to observe whether the sliding clamp proteins were conserved in their respective domain, a phylogenetic tree was constructed for different domains. The first step in this process was the sequence retrieval.

From the three domains of life, one organism each was chosen and their protein sequence for Beta-Clamp or PCNA was downloaded. The beta-clamp protein from *E. coli* (Uniport Accession Number: **P0A988)** was PSI-blasted against Non-Redundant Protein Sequences (NRPS) of 75 different species in bacterial domain, and all homologs of the protein from different organisms were downloaded. Similarly, the PCNA from *Pyrococcus furiosus* (Uniport Accession Number: **073947)** was PSI-blasted against Non-Redundant Protein Sequences of 74 archaeal species and PCNA protein from *Homo sapiens* (Uniport Accession Number**: P12004)** was PSI-blasted against Non-Redundant Protein Sequences of 31 eukaryal species [15] to obtain homologous proteins in each domain. The organisms in bacterial and archaeal domain were chosen from a list of organisms used by Moreira in 2014 [16]. The organisms were selected to represent different classes of the domains, and also had their whole genome sequenced.

The obtained protein sequences were then aligned using Muscle algorithm [17] with default settings in MEGA X. In this way, different Multiple Sequence Alignments were generated for the three domains of life.

## Tree construction

The evolutionary history was inferred by using the Maximum Likelihood method and JTT matrix-based model [18]. The bootstrap consensus tree inferred from 500 replicates was taken

to represent the evolutionary history of the taxa analyzed . Branches corresponding to partitions reproduced in less than 50% bootstrap replicates were collapsed. The percentage of replicate trees in which the associated taxa clustered together in the bootstrap test (500 replicates) were shown next to the branches [19]. Initial tree(s) for the heuristic search were obtained automatically by applying Neighbor-Join and BioNJ algorithms to a matrix of pairwise distances estimated using a JTT model, and then selecting the topology with superior log likelihood value. All positions with less than 95% site coverage were eliminated, i.e., fewer than 5% alignment gaps, missing data, and ambiguous bases were allowed at any position (partial deletion option). The evolutionary analyses were conducted in MEGA-X [20].

## Construction of tree of life

A phylogenetic tree of life was constructed from all three domains using selected species (selected at random). A total of 17 eukaryotes, 16 bacteria and 18 archaea species (selected at random representing different classes) were used to construct a phylogenetic tree of life. Only the second and third domains of bacterial species Beta Clamp (identified from Pfam) were used for tree construction. The evolutionary history was inferred using Maximum Likelihood method and JTT matrix-based model with 1000 bootstraps. All the other parameters and analysis steps were the same as in the phylogenetic tree of different domains of life.

## Result and discussion

### Detection of homology between PCNA and sliding clamp

With a threshold of 0.005, *P. furiosus* PCNA was PSI blasted against the NCBI non-redundant bacterial protein sequence database to the second iteration. Fig 1A shows a screenshot of the top hits from the blast. A number of PCNAs were discovered in the bacterial genome. Given that PCNAs are characteristic proteins of eukaryotes and archaea, these proteins may have arisen by horizontal gene transfer from eukaryotes or archaea to bacteria. Further analysis is necessary to determine their source of origin. The proteins not found in the first iteration, but found in the second iteration are highlighted in yellow as in Fig 1B. A slew of bacterial beta clamps were discovered as homologues of *P. furiosus* PCNA upon PSI blast at second iteration. The top among the hits was *Kallipyga gabonensis* beta clamp. The query cover for the blast was 94 percent, E score was $1 \times 10^{-4}$, and sequence identity was above 19%. Since a random blast of PCNA yielded beta clamps of various species, it can be claimed that the two are homologous.

A closer inspection of the alignment result shows that out of the three domains of beta clamp, domains 2 and 3 align with the full length of the PCNA. This suggests that either the Lowest Universal Common Ancestor (LUCA) had three domains in its sliding clamp, the first of which got lost in archaea, or LUCA had two domains in its sliding clamp and the third domain got added after a duplication event. The alignment of *K. gabonesis* beta clamp and *P. furiosus* PCNA shows 20% (50/249) sequence identity, 44% (109/249) sequence similarity and 11% (27/249) gaps. On the other hand, *P. furiosus* PCNA and human PCNA show 25% (61/249) sequence identity, 61% (136/249) sequence similarity and 5% (13/249) gaps. *E. coli* beta clamp and human PCNA, which were earlier claimed to not have sequence homology, show 16% (42/261) sequence identity, 36% (94/261) sequence similarity and 16% (43/261) gap. The alignments can be observed in Fig 2. These results demonstrate that although the more common model organism (human and *E. coli*) PCNA and beta clamp do not demonstrate high degree of homology, other organism (*K. gabonesis* and *P. furiosus*) beta clamp and PCNA demonstrate statistically noteworthy sequence homology.

A.

| Description | Max score | Total score | Query cover | E value | Per. Ident | Accession |
|---|---|---|---|---|---|---|
| proliferating cell nuclear antigen (pcna) [Flavobacteriales bacterium] | 222 | 222 | 96% | 1e-68 | 23.74% | MAD50601.1 |
| proliferating cell nuclear antigen (pcna) [Acidimicrobiaceae bacterium] | 218 | 218 | 98% | 2e-67 | 23.32% | MBG02288.1 |
| proliferating cell nuclear antigen (pcna) [Dehalococcoidaceae bacterium] | 217 | 217 | 99% | 3e-67 | 30.92% | MBR74595.1 |
| proliferating cell nuclear antigen (pcna) [Alphaproteobacteria bacterium TMED62] | 217 | 217 | 96% | 1e-66 | 21.76% | OUU62278.1 |
| proliferating cell nuclear antigen (pcna) [Halobacteriovoraceae bacterium] | 217 | 217 | 96% | 1e-66 | 23.55% | MAW08577.1 |
| proliferating cell nuclear antigen (pcna) [Flavobacteriales bacterium TMED191] | 217 | 217 | 96% | 1e-66 | 23.74% | RPG58329.1 |
| proliferating cell nuclear antigen (pcna) [Candidatus Marinimicrobia bacterium] | 216 | 216 | 97% | 1e-66 | 21.12% | MBD23483.1 |
| proliferating cell nuclear antigen (pcna) [Flavobacteriaceae bacterium] | 217 | 217 | 96% | 1e-66 | 23.14% | MAD12048.1 |
| proliferating cell nuclear antigen (pcna) [Nitrospinae bacterium] | 216 | 216 | 97% | 2e-66 | 21.83% | MBE18271.1 |

B.

| | | | | | | |
|---|---|---|---|---|---|---|
| DNA polymerase III subunit beta [Kallipyga gabonensis] | 53.8 | 53.8 | 94% | 1e-04 | 19.34% | WP_053942720.1 |
| hypothetical protein DRP84_07425 [Spirochaetes bacterium] | 52.7 | 52.7 | 94% | 2e-04 | 18.70% | RKX94156.1 |
| DNA polymerase III subunit beta [Kallipyga massiliensis] | 51.5 | 51.5 | 94% | 6e-04 | 18.11% | WP_019133621.1 |
| DNA polymerase III subunit beta [Cyanobacteria bacterium QH_2_48_84] | 50.0 | 50.0 | 92% | 0.002 | 16.67% | PSO65263.1 |
| DNA polymerase III subunit beta [Cyanobacteria bacterium QH_7_48_89] | 50.0 | 50.0 | 92% | 0.002 | 16.67% | PSO58692.1 |
| DNA polymerase III subunit beta [Cyanobacteria bacterium QH_1_48_107] | 49.6 | 49.6 | 92% | 0.003 | 16.67% | PSO55030.1 |
| DNA polymerase III beta subunit [Edaphobacter modestus] | 49.6 | 49.6 | 87% | 0.003 | 13.87% | RZU41692.1 |

**Fig 1. Screenshot of the results obtained when *P. furiosus* PCNA is PSI blasted against the NCBI bacterial database.** (A) shows the results obtained when regular blast is carried out. PCNA sequences in undefined bacterial genomes appear. It is possible that these bacterial PCNAs emerged in the bacterial genome from horizontal gene transfer from archaea and eukarya. (B) shows bacterial beta clamp from some defined and some undefined species that appear in the blast after the second iteration. The yellow color represents hits that only appear in the second iteration of the blast. *Kallipyga gabonensis* is the top defined beta clamp hit in the bacteria. Other defined bacterial beta clamps that appear in the blast search are *Kallipyga massiliensis* and *Edaphobacter modestus* beta clamps. Query cover above 90 percent in these searches indicates that the entire PCNA was covered during the blast search.

Given that domain 2 and 3 of beta clamp align with the two domains of PCNA, it can be hypothesized that domain 1 from beta clamp interacts with bacteria specific replication and repair protein. Most of the interaction assays carried out with PCNA and beta clamp have been conducted with the entire protein. It would be interesting to zero in on which domain of the sliding clamps is important for individual interactions.

Fig 2(A) and 2(D) represents the secondary structures found in beta clamps and PCNA. As we can see in the two figures, the secondary structures in both PCNA and beta-clamps are almost similar and conserved in all regions. Preservation of those secondary structures is essential for structural and functional conservation. Thus, despite low sequence identity between beta-clamps and PCNA, the secondary structures are conserved in both of them, causing structural and functional similarity.

## Phylogenetic tree of eukaryotic PCNA

To study the origin and evolution of eukaryotic PCNA, PCNA from humans was protein blasted in NCBI against the non-redundant protein sequences of selected eukaryotic organisms from different classes. Certain species of the kingdom/class eozoa, amoebozoa, ciliophora, archaeplastida, fungi and animalia were chosen. Where multiple PCNAs were discovered, all sequences were downloaded. A phylogenetic maximum likelihood tree was drawn using these sequences. A tree thus constructed is shown in Fig 3. The proteins from different classes tended to segregate as labeled on the right. For example, all PCNAs from ciliophora and animalia cluster together. This demonstrates that there is very little horizontal gene transfer. A largely vertical inheritance pattern can be expected for an essential protein like PCNA. However, a number of species in ciliophora class demonstrate gene duplication events. Three PCNA genes are found in both *Stylonychia lemnae* and *Oxytricha trifallax*. Each of the three

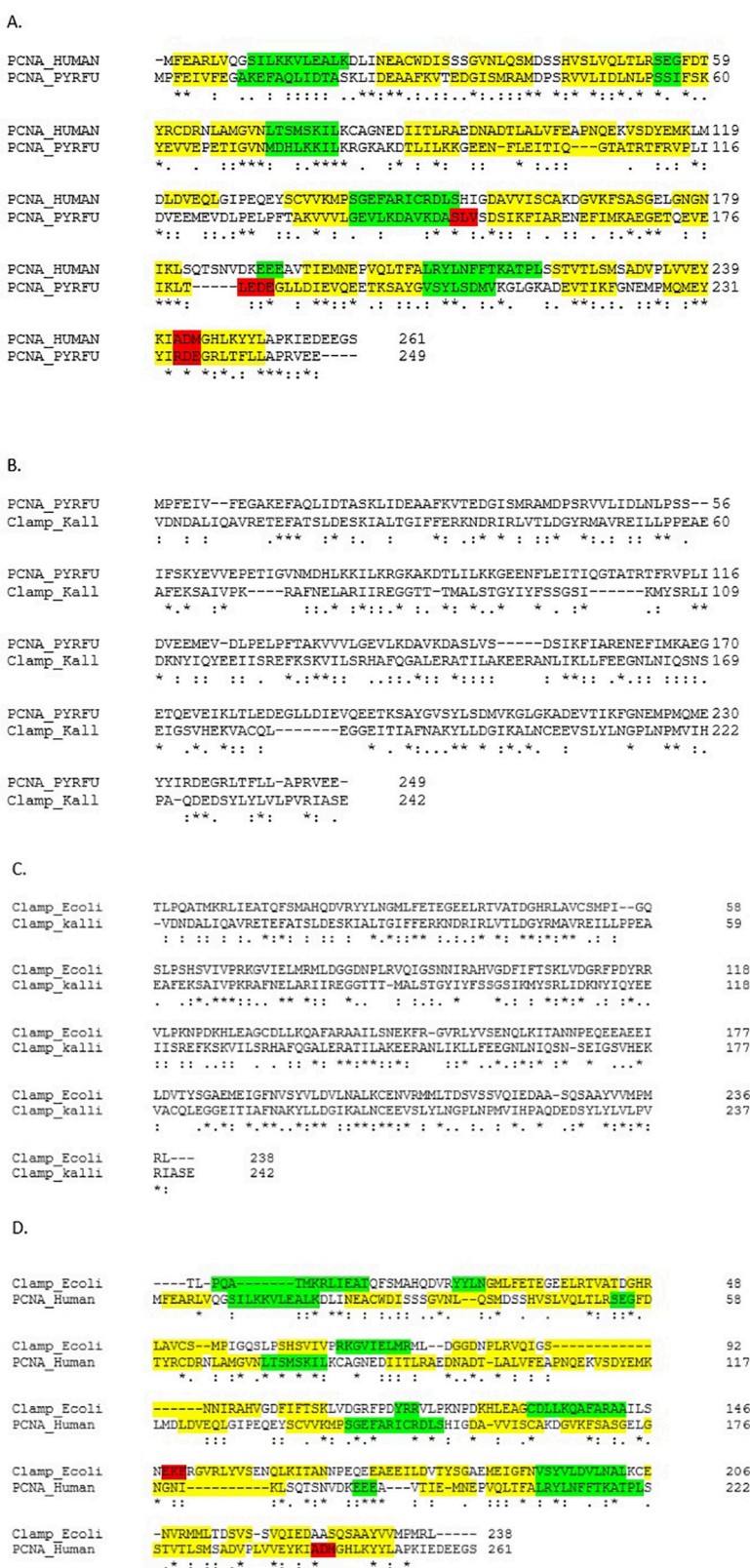

**Fig 2.** Clustal omega alignment of (A) Human and *P. furiosus* PCNAs (B) *P. furiosus* PCNA and *K. gabonensis* beta clamp (C) *E. coli* beta clamp and *K. gabonensis* beta clamp (D) *E coli* beta clamp and human PCNA. When PCNA and beta clamps were aligned, domain 1 of PCNA was aligned to domain 2 of beta clamp and domain 2 of PCNA was aligned to domain 3 of beta clamp. Since (A) and (C) align PCNA to PCNA and beta clamp to beta clamp (domains 2 and 3 aligned), they show the highest degree of similarity and identity. (B) *P. furiosus* PCNA and *K. gabonensis* beta

clamp also display a high degree of sequence identity and sequence similarity that is statistically significant. In the figure * represents sequence identity,: represents high sequence similarity (similar category of amino acids) and. represents low sequence similarity. (A) and (D): Color represents the secondary structures present in beta clamps of *E. coli* and PCNA of Human and *P. furiosus*. Yellow represents beta sheets, green represents helix and red represents turns.

proteins from one species lies next to a protein from another species. This suggests that a single PCNA gene underwent gene duplication twice in the progenitor organism, which gave rise to three PCNAs, each of which evolved separately when speciation occurred.

The figure also demonstrates that there are four closely related PCNAs in *Paramecium tetraurelia*. The tree pattern suggests that there was an initial duplication event, followed by duplications of the duplicated PCNA.

Amoebozoa and eozoa PCNAs tended to cluster at two locations each. *Trichomonas vaginalis* had two PCNAs, one clustering with other eozoa PCNAs and other clustering with

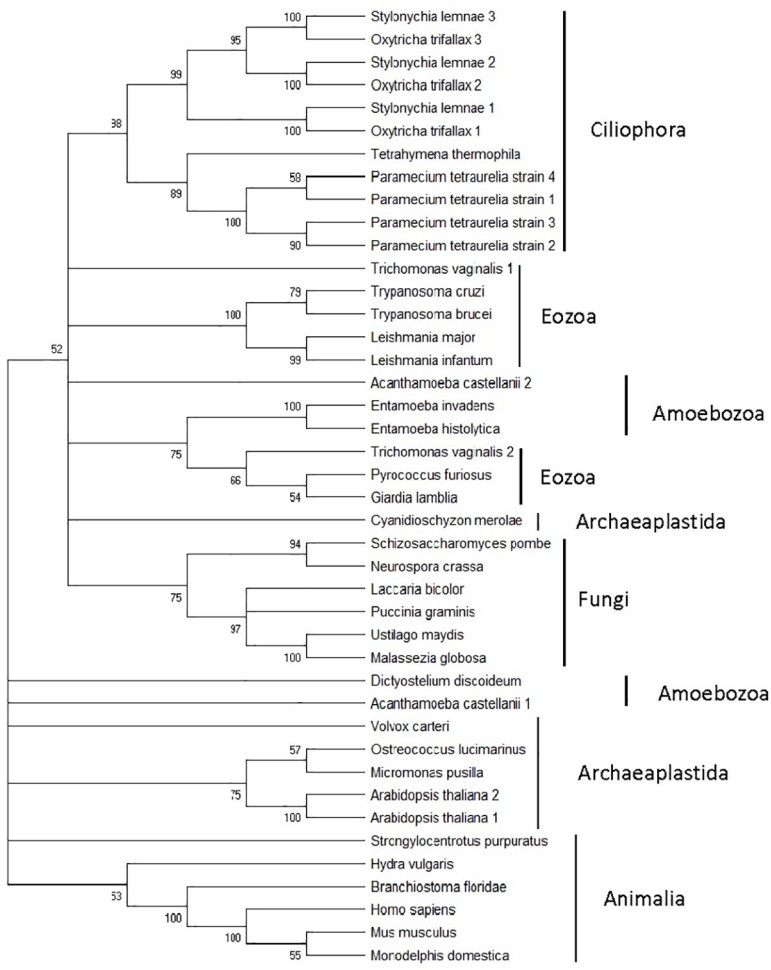

**Fig 3. Phylogenetic maximum likelihood tree drawn for PCNA of 31 different eukaryotic species.** Both domains of PCNA were used for alignment. Species from 6 kingdoms of eukaryotes—animalia, fungi, archaeaplastida, amoebozoa, eozoa and ciliophora (inside the kingdom chromista)—were used to draw the tree. The purpose of using organisms from 6 different kingdoms is to represent the entire eukaryotic domain, not just the crown groups. This tree demonstrates that most species contain only one PCNA that is vertically inherited. The instances where there is gene duplication and horizontal gene transfer is indicated in the text.

*Pyrococcus furiosa* (archaea) PCNA. This demonstrates that one PCNA was vertically inherited, while other PCNA might have arisen from horizontal gene transfer from archaea. Other eozoa (*Leishmania* and *Trypanosoma*) PCNA clustered together, suggesting that they all came from a common ancestor. *Giardia lambia* (an eozoa) PCNA, however segregated separately close to *Pyrococcus furiosus* PCNA, hinting at a possible archaeal origin.

As for amoebozoa PCNAs, three PCNAs from different species lie next to each other, whereas two other PCNAs, one from *Dictostylium discoideum* and another from *Acanthamoeba castellani*, are sandwiched between archeaplastida and fungi. It has been observed from earlier research that *Dictostylium* molecules resemble archeaplastida and fungi molecules more than amoebozoa molecules [21, 22]. So, this placement in the tree does not come as a surprise. The second *Acanthamoeba castellani* PCNA might have arisen from horizontal gene transfer from *Dictosylium* or archaeaplastida or fungi.

All fungi and animalia PCNAs cluster together suggesting a common origin. There seems to be no gene duplication event in the species we have considered in the paper. All except for one, archaeplastida molecules cluster together also suggesting a common origin. *Cyanidioschyzon merolea* PCNA is found at a different location from other archaeplastida genomes suggesting a different ancestry.

## Phylogenetic tree of bacterial beta clamp

The beta-clamp from *E. coli* was PSI-blasted against non-redundant protein sequences from different organisms belonging to different phyla in bacteria. A total of 85 homologs of the protein were identified from 75 species of bacteria. All the homologs were downloaded, and aligned together to build a phylogenetic tree as shown in Fig 4. It was observed that most of the bacteria consisted of only one homolog for beta-clamp. However, some bacteria had two or more homologs of the protein, suggesting ancient gene duplication or horizontal gene transfer. The bacteria belonging to the same phylum, or sharing similar properties were found to be clustered together apart from some exceptions, which suggests that the beta-clamp protein is conserved among similar species in the bacterial domain.

In proteobacteria, three classes viz. Alphaproteobacteria, Betaproteobacteria and Gammaproteobacteria are found to clustered together, originating from a single node suggesting a common origin. However, the Delta/Epsilon sub-divisions seem to scatter away from other Proteobacteria in the phylogenetic tree. It suggests that the phylum Proteobacteria is not monophyletic which has also been previously suggested [23]. The phyla Planctomycetes, Verrucomicrobia and Chlamydiae which are often grouped together as PVC superphylum [23] were observed to be clustered together, suggesting similar origin and properties.

Multiple homologs of the same protein in a species can be accounted for by ancient gene duplication or horizontal gene transfer events. In the case of *Oscillatoria nigroviridis*, four homologs of sliding beta clamp were observed. It was a result of three ancient gene duplications at different stages of time, followed by vertical gene transfer. Similar was the case for *Bacillus thirungenesis*. The resulting homologs did not undergo much variation and thus were clustered together. The phylum Actinobacteria was positioned at two different locations in the phylogenetic tree, one clustered with Cyanobacteria and the other with Deinococcus-thermus. It suggests that the horizontal gene transfer occurred from the ancestor of Deinococcus-thermus to actinobacteria before speciation. This gene was maintained in some species, which showed two homologs for beta-clamp protein, while it has disappeared in some species (*Nocardia brasiliensis*), which only had a single homolog.

In the phylum Fusobacteria, two homologs each of *Ilyobacter polytropus* and *Fusobacterium nucleatum subsp. nucleatum* were observed, which were clustered together. It might have

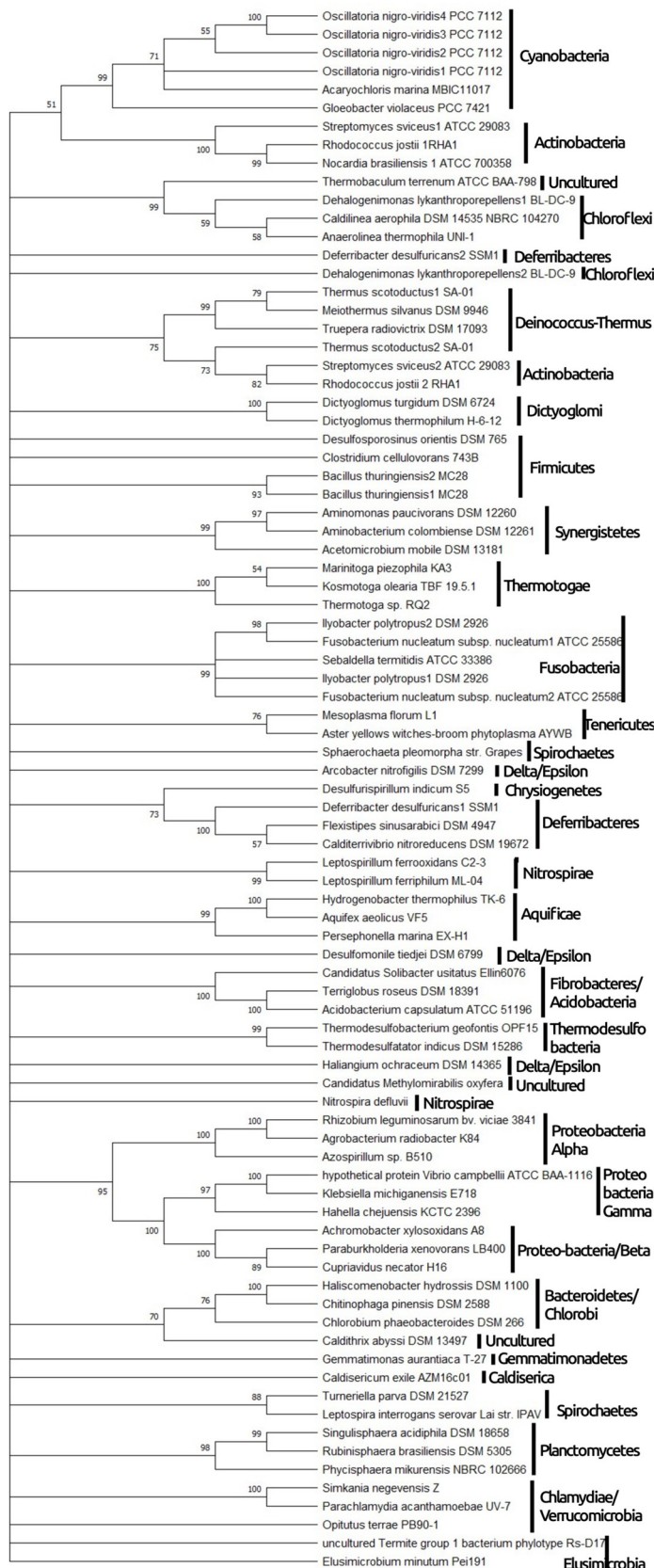

**Fig 4. Maximum likelihood tree drawn for beta-clamp of 75 different bacterial species representing different phylum and classes.** All three domains of beta-clamp were used for alignment. The tree demonstrates that most bacteria have only a single homolog for beta-clamp protein, which is vertically inherited. Some instances of gene duplication and horizontal gene transfer are also observed. Bacteria belonging to the same class are clustered together with minor exceptions.

resulted from ancient gene duplication, or lateral gene transfer among the species. Two homologs of the beta-clamp protein of *Deferribacter desulfuricans* were observed, where one homolog was separated from its counterpart and located between species of Chloroflexi suggesting lateral gene transfer from Chloroflexi. Other cases of lateral gene transfer can be claimed in *Nitrospira defluvii*, and *Dehalogenimonas lykantroporepellens*.

Apart from some cases of horizontal gene transfer and gene duplication, the species from the same class were clustered together in the phylogenetic tree. Thus, we can conclude that beta-clamp protein is fairly conserved in the bacterial domain.

## Phylogenetic tree of archaeal PCNA

The PCNA from *P. furiosus* was PSI-blasted against non-redundant protein sequences from different archaea belonging to different classes in archaea. 115 homologs of PCNA were identified in 74 species of archaea. All of the homolog sequences were downloaded, and a multiple sequence alignment and phylogenetic tree was constructed similarly as discussed above.

The obtained phylogenetic tree is shown in Fig 5. It can be observed that the three main phyla of domain archaea, viz Thaumarchaeota, Euryarchaeota and Crenarchaeota were positioned separately in the phylogenetic tree, apart from some exceptions. The phylum Thaumarchaeota was sandwiched in between Euryarchaeota and Crenarchaeota. A single species of Korachaeota was present within the phylum Crenarchaeota, which leads to question the monophyly of Crenarchaeota.

In the phylum, Euryarchaeota, the species belonging to the same class were classified together. However, some cases of duplication and horizontal gene transfer were observed. Two homologs of PCNA were observed in *Halogeometricum borinquense DSM 11551*, which were separated from the same node, hence suggesting ancient gene duplication. Similar was the case for *Methanotorris igneus Kol5*. Also, two homologs of the protein were discovered for *Haladaptatus paucihalophilus*, and *Canditatus Nanosalina*. These two homologs were not positioned together, hence suggests horizontal gene transfer.

The phylum Crenarchaeota is divided into three classes, viz. Sulfolobales, Desulfurococcales, and Thermoproteales. The phylogenetic tree obtained still supports the close relationship of Sulfolobales and Desulfurococcales, also mentioned by Armanet [24]. Both these classes had three homologs of PCNA. The two homologs appear to have originated by gene duplication in the ancestor of these two classes, followed by modification to form different proteins, while the third homolog might have been acquired by lateral gene transfer from other diverse species. As all three homologs are required for the assembly of a functional hetero-trimer, the homologs are observed to be vertically inherited in future generations. The hetero trimeric PCNA of *Sulfolobus solfataricus* can be hypothesized to have origins described for Sulfolobales and Desulfurococcales. However, the third phylum of Crenarchaeota: Thermproteales had two homologs of PCNA. Thus, it can be predicted that the functional PCNA in Thermoproteales is a homo-trimer (of any one of the homolog) and two homologs might have arisen due to ancient gene duplication.

The phylum Euryarchaeota can be divided into two groups: Sub-Phyla I and Sub-Phyla II. Sub-Phyla I consists of Thermcoccales, Nanoarchea, and class I methanogens, while Sub-Phyla II consists of Archeoglobulus, Halobacteriales, Thermoplasmatales and Class II methanogens

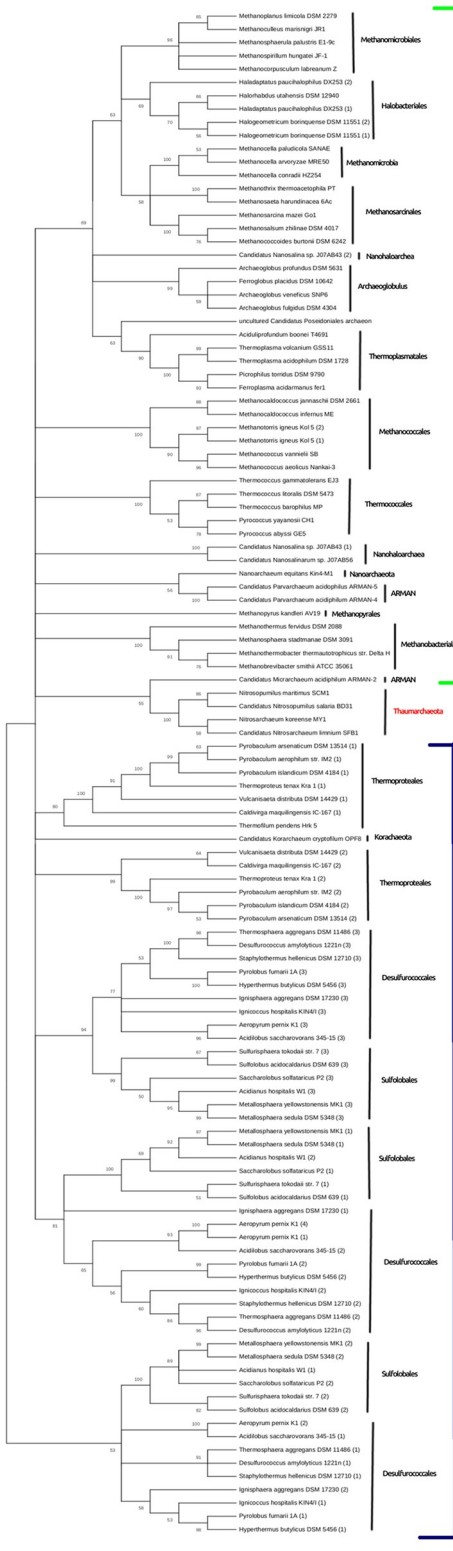

**Fig 5. Maximum likelihood tree drawn for PCNA of 75 different archaeal species representing three major phyla: Euryarchaeota, Thaumarchaeota and Crenarchaeota (green line: Euryarchaeota, blue line: Crenarchaeota).** The tree demonstrates that the three phyla are positioned separately. Some archaea are observed to have more than one homolog for PCNA. It is observed that PCNA protein is highly conserved in archaea, with few exceptions of gene duplication and horizontal gene transfer.

[25]. The phylogenetic tree in Fig 5 suggests the monophyly of Sub-Phyla II, which is also supported by the updated tree of life by Forteree [25]. However, the monoplyly of Sub-Phyla I could not be supported statistically.

The phylogenetic tree further clarifies the position of Methanopyrales in the phylogenetic tree. As *Methanopyrales kandleri* was positioned along with Methanobacteriales, Methanopyrales can be classified as Class I Methanogen. However, the monophyly of Class I and Class II methanogen could not be supported. Methanobacteria, Methanobacteriales and Methanosarcinales were observed to share a common ancestor along with Halobacteriales. Methanococcus was positioned near to Thermococcales, while Nanohaloarchea and Nanoarchaeota were sandwiched between Methanococcus and Methanobacteriales.

The phyum Nanoarchaeota was positioned near Thermococcales, along with Nanohaloarchaea, and the ARMAN group. The close association of Nanoarchaeota to Thermococcales was also observed by Brochier in 2005 [26]. Also, Thermococcales, Nanoarchaea, ARMAN and Nanohaloarchea are placed together suggesting they are closely related. It has also been supported by phylogeny based on ribosomal proteins and DNA replication proteins [25, 27]. The tree, however, questions the position of ARMAN-2. It has been positioned as a sister group to Thaumarchaeota, forming a border-line between Crenearchaeota and Euryarchaeota.

Thus, the phylogenetic tree based on archaeal PCNA follows a vertical pattern of evolution in most cases. Also, archaeal PCNA is conserved among closely related species of the Archaea domain.

## Phylogenetic tree of life

A total of 61 protein sequences (obtained from 17 eukaryotes, 16 bacteria and 18 different archaea species) selected at random representing different classes were used to construct a phylogenetic tree of life. Only the second and third domain of bacterial beta clamp was observed to align with PCNA protein. Thus, The bacterial sequences were processed using Pfam [28] website and sequences spanning second and third domain were used for tree construction. A tree thus constructed is shown in Fig 6.

As observed from that figure, all eukaryotes and all bacterial species have descended from a single but separate node, while archaeal species are scattered throughout the phylogenetic tree. This suggests that beta clamp protein is fairly conserved across the bacterial domain, and PCNA is highly conserved among the eukaryotes. But, PCNA is archaea has sufficiently diverged among different species while maintaining common structure and functionality. Since bacteria, archaea and eukarya form different clusters in the phylogenetic tree, we can infer that although sliding clamp protein might have had a single origin in the past, the proteins have sufficiently diverged across different domains of life remaining fairly conserved in its own domain.

## How does PCNA and beta-clamp maintain common structure and functionality?

Although PCNA and Beta-Clamp do not share prominent sequence identity, they have the same structure and perform the same function. Both PCNA and beta-clamp might have had some conserved residues that resulted in the same function. In order to identify such

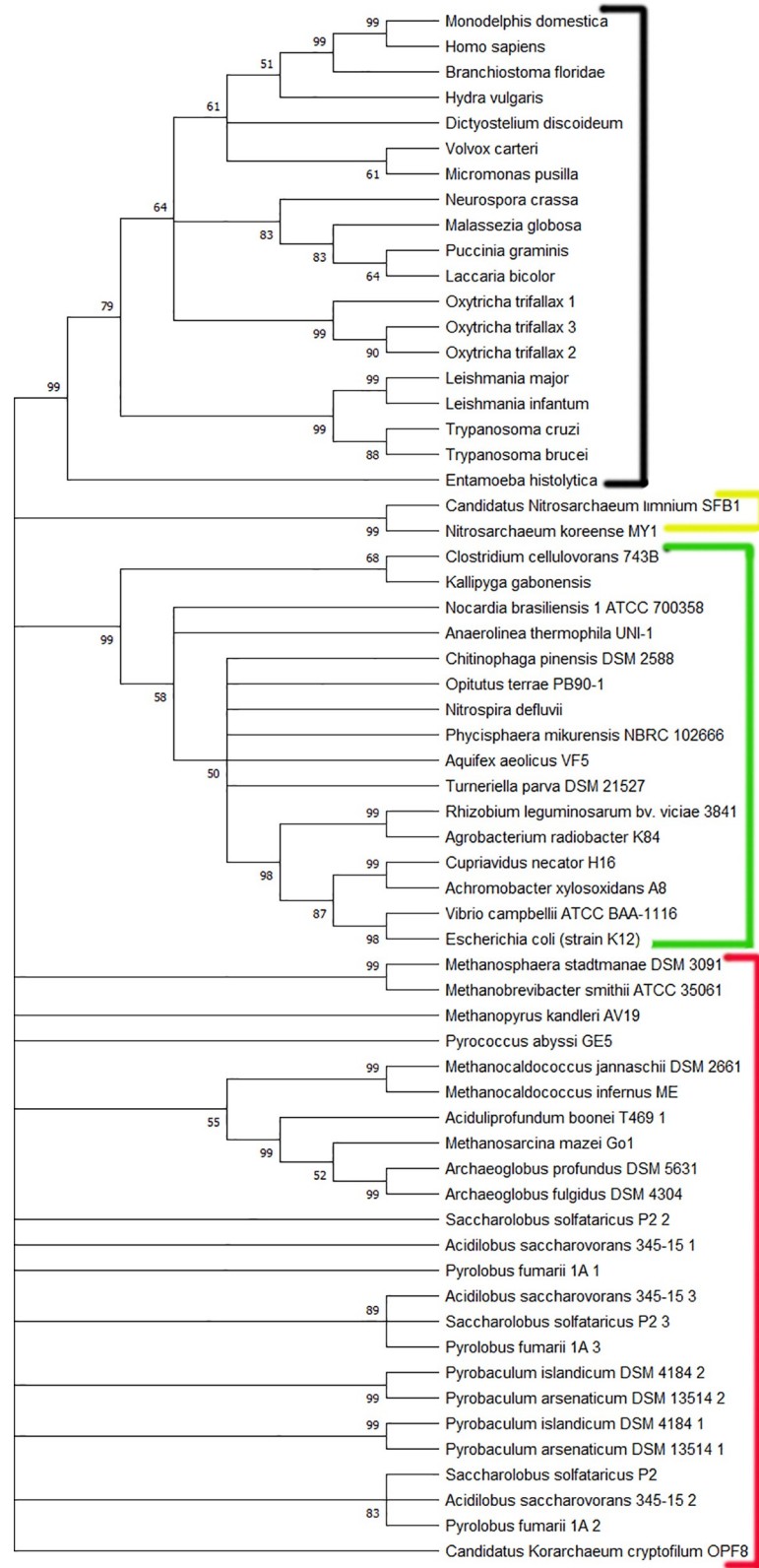

**Fig 6. A phylogenetic tree of 61 amino acid sequences representing species from all domains of life.** Black line represents Eukarya domain, Green line represents Bacteria domain, Yellow line represents Korchaeota, and Red line represents Archaea domain. It can be observed that all bacterial species originate from a single node, and all eukaryal species descended from a single, but separate node. However, archaeal species are separated throughout the phylogenetic tree.

conserved residues, we did a multiple alignment of sequences of sliding clamp proteins in all three domains of life separately using CLUSTAL-OMEGA. In total, we found 18 conserved residues (at >60% conservation threshold) in archaea (as shown in S3 Fig) and 69 conserved residues (at >80% conservation threshold) in eukarya (can be observed in S1 Fig). In bacteria, when all three domains of beta-clamp were selected for alignment, only 23 residues were conserved at 60% conservation level (as shown in S2 Fig). But, when only domains 2 and 3 of all beta-clamp sequences were aligned, 27 residues were observed to be conserved at 60% conservation threshold (as shown in S5 Fig). The higher conservation in domain 2 and 3 of beta-clamps puts forward the possibility of an important structural and functional role of the second and third domain.

Next, we did a multiple alignment of species from all three domains of life (From bacteria, only domains 2 and 3 of beta-clamp were selected for alignment) using CLUSTAL-OMEGA tool. The alignment figure is available in supplementary information (S4 Fig). A total of 10 residues were found to be conserved across all domains of life at 60% conservation level. Among them, 8 residues each were conserved in Eukarya, 7 in Archaea, while 6 residues were conserved in Bacterial domain. We have used low conservation threshold to identify many residues which could potentially have some functionality in sliding clamp proteins. Since our conservation threshold is around 60%, it is possible that some of the residues might be conserved in only two of the three domains, but still identify as conserved in inter-domain alignment. We did a multiple alignment of Archaea and Eukarya Domain (all PCNAs) to further explore this argument (as shown in S6 Fig). All residues of concern (not conserved in Bacteria) were found to be conserved in a multiple alignment of PCNAs. In addition, it explains all unconserved residues in Archaea and Eukarya. As observed in the phylogenetic tree of life, Archaea domain remains scattered in the tree. Also, there is high variability in the PCNAs in the Archaeal domain (comprising of hetero-trimers and homo-trimers). It is highly possible that a group of archaeal species (having homo-trimer PCNAs) might share more similarity with eukaryal species (which also have homo-trimer PCNAs) than fellow archaea. This can result in the identification of new conserved residues not observed in the respective domains. Further study of these conserved residues can shed new light in the evolution of sliding clamp proteins across different domains of life.

To conclude, the second and third domain of beta-clamps share some sort of sequence homology with PCNA proteins. In addition, the conservation of secondary structures in second and third domain of beta clamps with PCNA proteins adds evidence to the hypothesis that LUCA had three domains in its sliding clamp, the first of which got lost in archaea, or LUCA had two domains in its sliding clamp and the third domain got added after duplication event. Furthermore, the sliding clamp proteins have been conserved throughout the different domains of life, thus preserving the replicative procedure and mechanism.

## Supporting information

**S1 Table. List of species from three domains of life used in this work.**
(PDF)

**S1 Fig. Multiple alignment of eukaryal PCNA under 80% conservation.** The highlighted residues are conserved at 80% conservation level.
(SVG)

**S2 Fig. Multiple alignment of bacterial beta-clamps under 60% conservation.** Only the highlighted residues are conserved at 60% cutoff.
(SVG)

**S3 Fig. Multiple alignment of archaeal PCNA under 60% conservation.** Only the highlighted residues are conserved at 60% cutoff.
(SVG)

**S4 Fig. Multiple alignment of sliding clamp proteins from all three domains of life under 60% conservation.** The sequences of sliding clamps from different species representing all domains and most classes were aligned and the highlighted residues depict the conserved residues across all three domains of life. Note that only the second and third domain of bacterial beta clamp are selected for alignment.
(SVG)

**S5 Fig. Multiple alignment of bacterial beta-clamps (second and third domain).** The highlighted residues are conserved at 60% conservation level.
(SVG)

**S6 Fig. Multiple alignment of archaeal and eukaryal PCNA.** The highlighted residues are conserved at 60% conservation level.
(SVG)

**S1 Data.**
(ZIP)

## Acknowledgments

We would like to express our gratitude to Dr. Anusha Thapa for thoroughly copyediting our manuscript.

## Author Contributions

**Conceptualization:** Hitesh Kumar Bhattarai.

**Data curation:** Sandesh Acharya, Hitesh Kumar Bhattarai.

**Formal analysis:** Sandesh Acharya, Amol Dahal, Hitesh Kumar Bhattarai.

**Investigation:** Sandesh Acharya, Amol Dahal.

**Methodology:** Sandesh Acharya.

**Project administration:** Hitesh Kumar Bhattarai.

**Resources:** Sandesh Acharya.

**Supervision:** Hitesh Kumar Bhattarai.

**Writing – original draft:** Sandesh Acharya, Amol Dahal, Hitesh Kumar Bhattarai.

**Writing – review & editing:** Sandesh Acharya, Amol Dahal, Hitesh Kumar Bhattarai.

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
