## [Decision Letter · Decision Letter 0]

15 Dec 2020

PONE-D-20-31084

Evolution and origin of sliding clamp in bacteria, archaea and eukarya

PLOS ONE

Dear Dr. Bhattarai,

Thank you for submitting your manuscript to PLOS ONE. Our apologizes for the length of time involved in this review, but individuals to evaluate the manuscript were scace in the pandemic climate. After careful consideration, we feel that it has merit but does not fully meet PLOS ONE’s publication criteria as it currently stands. Therefore, we invite you to submit a revised version of the manuscript that addresses the points raised during the review process.

The issues surrounding this manuscript are methodological, data transparency, and the best means for interpretation of the data. These issues relate to the need to analyze the data and provide rigorous interpretation of the results. The following changes need to be made within the text of the manuscript.Provide the methodology for the second iteration of PSI-BLAST.Provide the full BLAST results possibly as a hyperlink.Discuss the phylogenetic and evolutionary analysis of the data suggested by Reviewer 2.Discuss the structural comparisons, also suggested by Reviewer 2.There are no conflicts between the two reviews.I have analyzed the manuscript and concur with Reviewer 2's evaluation/

We look forward to receiving your revised manuscript.

Kind regards,

Arthur J. Lustig, PhD

Academic Editor

PLOS ONE

Journal Requirements:

2.We suggest you thoroughly copyedit your manuscript for language usage, spelling, and grammar. If you do not know anyone who can help you do this, you may wish to consider employing a professional scientific editing service.  

3. Please ensure that you refer to Figure 2 in your text as, if accepted, production will need this reference to link the reader to the figure.

Reviewers' comments:

Reviewer's Responses to Questions

**Comments to the Author**

1. Is the manuscript technically sound, and do the data support the conclusions?

Reviewer #1: Yes

Reviewer #2: Partly

2. Has the statistical analysis been performed appropriately and rigorously? 

Reviewer #1: N/A

Reviewer #2: N/A

3. Have the authors made all data underlying the findings in their manuscript fully available?

Reviewer #1: Yes

Reviewer #2: No

4. Is the manuscript presented in an intelligible fashion and written in standard English?

Reviewer #1: Yes

Reviewer #2: Yes

5. Review Comments to the Author

Reviewer #1: If possible, the manuscript and field would benefit from utilizing the current approach to analyze, interpret and/or discuss organisms, such as the arcaeaon Sulfolobus solfataricus P2, that contain heterotrimeric PCNA sliding clamps. How did each subunit evolve, etc.?

Reviewer #2: This study conducted sequence similarity analysis of archaeal and eukaryotic PCNAs and their bacterial counterpart beta clamps. They showed that shows PCNA likely has some sequence homology with beta clamp, contradicting previous claims. They further built phylogenetic trees within each kingdoms of life. This information could be useful reference in the field of DNA replication. The manuscript can be significantly improved if more information is provide as suggested below.

Major comments:

1. It is unclear how the authors conducted the second iteration of PSI-blast.

2. The authors need to provide the full blast results, perhaps as supplementary information.

3. It is unclear whether the low sequence conservation between, for instance, Fungal PCNA and bacterial beta clamp, is functionally important. It would be informative if the authors can score each residue for their conservativeness within each kingdom of life. This way, they can test whether the more conserved residues within each kingdom are more likely to be conserved across kingdoms.

4. To help readers understand the sequence conservation in relation to structure, the authors could provide domain information along with the alignment.

5. Is it possible to build a tree from selective species from all three kingdoms of life?

Minor Comments:

1. There should be spacing between a word and a citation. There should be no spacing between a citation and a period.

2. There is no labeling for Figure 1A.

6. PLOS authors have the option to publish the peer review history of their article (what does this mean?). If published, this will include your full peer review and any attached files.

Reviewer #1: No

Reviewer #2: No

---

## [Author Response · Author response to Decision Letter 0]

29 Jan 2021

We would like to extend our gratitude to both our reviewers for providing their valuable suggestions to make our paper more informative. Following the recommendations from both our reviewers, the following contents were added to the manuscript.

• The procedure for second iteration of PSI-Blast was provided. 

• Full Blast Results of the Second Iteration was provided as a hyperlink.

• Domain Information (Secondary Structures including helix, turns, sheets) was provided with the alignment to help the readers understand the conservation of secondary structures across the domains of life.

• A tree of life was constructed using sequences from all three domains of life. 

• From the multiple sequence alignment files of sliding clamp proteins of each domain, the residues which were conserved in the respective domain were identified. Furthermore, the multiple alignment of sequences across all domains of life was generated, and the residues which were conserved across all domains were identified and compared with the former. 

• The evolutionary history of Sulfolobus solfataricus was explained with reference to Sulfolobales and Desulfurococcales. 

Following the suggestion by the reviewers, the paper was sent for copyediting. The paper was copyedited by Dr. Anusha Thapa. Furthermore, the contents of the manuscript were formatted as per the PLOS One Author Guidelines.

---

## [Decision Letter · Decision Letter 1]

3 Mar 2021

PONE-D-20-31084R1

Evolution and origin of sliding clamp in bacteria, archaea and eukarya

PLOS ONE

Dear Dr. Bhattarai,

Thank you for submitting your manuscript to PLOS ONE. After careful consideration, we feel that it has merit but does not fully meet PLOS ONE’s publication criteria as it currently stands. Therefore, we invite you to submit a revised version of the manuscript that addresses the points raised during the review process.

The format of the rebuttal provided insufficient information for a reviewer to understand the textual modification made by the authors. Please rewrite this rebuttal in in a point-by-point fashion, specifically addressing the content and location of the text that was modified in the revision. It will then be re-evaluated.The issue of inter-kingdom versus intra-kingdom residue conservation must be clearly stated in the text and rebuttal.

We look forward to receiving your revised manuscript.

Kind regards,

Arthur J. Lustig, PhD

Academic Editor

PLOS ONE

Journal Requirements:

Reviewers' comments:

Reviewer's Responses to Questions

**Comments to the Author**

1. If the authors have adequately addressed your comments raised in a previous round of review and you feel that this manuscript is now acceptable for publication, you may indicate that here to bypass the “Comments to the Author” section, enter your conflict of interest statement in the “Confidential to Editor” section, and submit your "Accept" recommendation.

Reviewer #2: (No Response)

2. Is the manuscript technically sound, and do the data support the conclusions?

Reviewer #2: Yes

3. Has the statistical analysis been performed appropriately and rigorously? 

Reviewer #2: Yes

4. Have the authors made all data underlying the findings in their manuscript fully available?

Reviewer #2: Yes

5. Is the manuscript presented in an intelligible fashion and written in standard English?

Reviewer #2: Yes

6. Review Comments to the Author

Reviewer #2: The rebuttal letter is too simple and was laid out in a way that facilitate reviewers to evaluate the point-by-point response. However, I do appreciate that the authors did try to address all my comments. Yet they need to be more considerate in drafting their response where the answers/conlusions should be obvious in the letter or the authors should point the reviewers to where their corresponding revisions are in the manuscript. The authors did not do their job sufficiently in this regard.

I appreciate that the authors provided multiple alignments of PCNA and beta clamps. But they have not addressed the question "whether the more conserved residues within each kingdom are more likely to be conserved across kingdoms". At not in an obvious way to readers.

7. PLOS authors have the option to publish the peer review history of their article (what does this mean?). If published, this will include your full peer review and any attached files.

Reviewer #2: No

---

## [Author Response · Author response to Decision Letter 1]

17 Apr 2021

We have formatted our response to reviewers, manuscript and revised manuscript following your suggestions. Hoping for a positive response.

---

## [Decision Letter · Decision Letter 2]

18 May 2021

PONE-D-20-31084R2

Evolution and origin of sliding clamp in bacteria, archaea and eukarya

PLOS ONE

Dear Dr. Bhattarai,

Thank you for submitting your manuscript to PLOS ONE. After careful consideration, we feel that it has merit but does not fully meet PLOS ONE’s publication criteria as it currently stands. Therefore, we invite you to submit a revised version of the manuscript that addresses the points raised during the review process.

You have satisfied all of the critiques with the exception of one. Reviewer 2 properly states a confusion in the phrase "The residues conserved in the interdomain alignment were not conserved in the intradomain alignment. This could be because the residues initially conserved between the domains were not important in individual domains as domains diverged and were chosen not to be conserved." The logic of the sentence is unclear since it seems that intra-domain conservation must precede inter-domain  conservation, as the reviewer points. This issue must be addressed before acceptance.

We look forward to receiving your revised manuscript.

Kind regards,

Arthur J. Lustig, PhD

Academic Editor

PLOS ONE

Journal Requirements:

Reviewers' comments:

Reviewer's Responses to Questions

**Comments to the Author**

1. If the authors have adequately addressed your comments raised in a previous round of review and you feel that this manuscript is now acceptable for publication, you may indicate that here to bypass the “Comments to the Author” section, enter your conflict of interest statement in the “Confidential to Editor” section, and submit your "Accept" recommendation.

Reviewer #2: (No Response)

2. Is the manuscript technically sound, and do the data support the conclusions?

Reviewer #2: Yes

3. Has the statistical analysis been performed appropriately and rigorously? 

Reviewer #2: Yes

4. Have the authors made all data underlying the findings in their manuscript fully available?

Reviewer #2: Yes

5. Is the manuscript presented in an intelligible fashion and written in standard English?

Reviewer #2: Yes

6. Review Comments to the Author

Reviewer #2: The authors have addressed most of my previous comments. I have one remaining issue:

"The residues conserved in the interdomain alignment were not conserved in the intradomain alignment. This could be because the residues initially conserved between the domains were not important in individual domains as domains diverged and were chosen not to be conserved."

This confusion does not make much sense to me. They found 13 residues conserved across all domains but only 3 of them inside each domain without showing any of data. Intuitively, the prerequisite to interdomian conservation is that the residues are conserved by intradomain alignment. These indicated to me that they may not have done the analysis properly. They should provide alignments and highlight the locations of these residues of interest so that readers can evaluate the data. Also instead of only reporting the 3 residues that are stringently conserved within all three domains, they should also report the percentage of the 13 residues that are conserved within each of the domains. They should also discuss the potential structural-function significance of the conserved residues.

7. PLOS authors have the option to publish the peer review history of their article (what does this mean?). If published, this will include your full peer review and any attached files.

Reviewer #2: No

---

## [Author Response · Author response to Decision Letter 2]

28 Jun 2021

The notion that intradomain conservation must precede inter-domain conservation holds true when the conservation level is very high. In our case, the threshold conservation level is set comparatively low to identify many important residues that might serve some functionality in sliding clamp proteins. We have three domains of life: bacteria, archaea and eukarya in our inter-domain multiple alignment. Since our conservation threshold is around 60%, it is possible that some of the residues might be conserved in only two of the three domains, but still identify as conserved in inter-domain alignment. A multiple alignment between archaea and eukarya (all PCNAs) was done, and it was identified that all residues of concern (observed conserved in the inter-domain alignment, but unconserved in bacteria, archaea and eukarya) were conserved in multiple alignment of PCNAs. This issue, along with its biological significance is addressed in line no 530-543 of revised manuscript with track changes.

---

## [Editor Report · Decision Letter 3]

8 Jul 2021

Evolution and origin of sliding clamp in bacteria, archaea and eukarya

PONE-D-20-31084R3

Dear Dr. Bhattarai,

We’re pleased to inform you that your manuscript has been judged scientifically suitable for publication and will be formally accepted for publication once it meets all outstanding technical requirements.

Kind regards,

Arthur J. Lustig, PhD

Academic Editor

PLOS ONE

Additional Editor Comments (optional):

All criticisms have been addressed.
---

## [Editor Report · Acceptance letter]

13 Jul 2021

PONE-D-20-31084R3 

Evolution and origin of sliding clamp in bacteria, archaea and eukarya 

Dear Dr. Bhattarai:

I'm pleased to inform you that your manuscript has been deemed suitable for publication in PLOS ONE. Congratulations! Your manuscript is now with our production department. 

Kind regards, 

on behalf of

Dr. Arthur J. Lustig 

Academic Editor

PLOS ONE